# Biocontrol Potential of *Bacillus velezensis* RS65 Against *Phytophthora infestans*: A Sustainable Strategy for Managing Tomato Late Blight

**DOI:** 10.3390/microorganisms13030656

**Published:** 2025-03-14

**Authors:** Hasna Elhjouji, Redouan Qessaoui, Hafsa Houmairi, Khadija Dari, Bouchaib Bencharki, El Hassan Mayad, Hinde Aassila

**Affiliations:** 1Agri-Food and Health Laboratory, Faculty of Science & Technology, Hassan First University of Settat, Settat 26000, Moroccoa_nacera2@yahoo.fr (K.D.); bouchaib.bencharki@uhp.ac.ma (B.B.); hindehoney@gmail.com (H.A.); 2Regional Center of Agricultural Research of Agadir, National Institute of Agricultural Research (INRA), Avenue Ennasr, BP415 Rabat Principal, Rabat 10090, Morocco; 3Laboratory of Biotechnology and Valorization of Natural Resources, Faculty of Science of Agadir, Ibn Zohr University, Agadir 80000, Morocco; e.mayad@uiz.ac.ma

**Keywords:** rhizosphere isolates, biocontrol activity, tomato late blight

## Abstract

This study aimed to investigate the biocontrol activity of rhizosphere isolates against late blight disease of tomatoes caused by the fungus *Phytophthora infestans*. A total of 30 rhizospheric bacterial isolates were evaluated for their antagonistic activity against *P. infestans in vitro* and *in vivo*. The results demonstrated that among the 30 isolates tested, six (RS65, RP6, RS47, RS46, RP2, and RS61) exhibited a highly significant inhibitory effect (*p* < 0.001) on the mycelial growth of *P. infestans in vitro*, with the inhibition rate exceeding 67%. Among the isolates, RS65 exhibited the highest inhibition rate at 78.48%. For antagonistic mechanisms, the results demonstrated that the six isolates exhibited significant enzymatic activity, including proteolytic, lipolytic, and chitinolytic activity, as well as the production of HCN, cellulase, and pectinase. Isolate RS65, which showed the highest inhibition rate, was further evaluated under greenhouse conditions. This investigation revealed significant differences in the severity of late blight between the control and the RS65 treatment. The control showed a severity level of 31.26%, whereas the RS65 treatment achieved the lowest severity of 16.54%. Molecular identification results indicated that the RS65 isolate (accession numbers PV208381) is a *Bacillus* genus with 99% proximity to *Bacillus velezensis*. This finding suggests that the *Bacillus* RS65 treatment could provide effective protection against *P. infestans* infection in tomato plants. These findings highlight the potential of *Bacillus* RS65 as a biocontrol agent in integrated disease management for tomato late blight.

## 1. Introduction

Tomato (*Solanum lycopersicum* L.) is one of the most consumed vegetables in the world due to its nutritional characteristics and benefits to health [1]. In Morocco, tomatoes grown in greenhouses cover 14,861 ha, with 1,347,085 tons produced in 2019, 90% of which were produced in the Souss region [2]. They are considered an essential crop in the Souss region [3,4]. In 2023, production increased to 1,444,678.6 tons, with a yield of 98,444.9 Kg/ha [2]. Tomato cultivation has great economic importance internationally, with 8,397,414.3 tons of fresh tomatoes being exported in 2020, of which Moroccan export was 7%, generating USD 764,876,000.00 in revenue [2].

Tomato crops are constantly subjected to various biotic stresses that affect their growth and limit agricultural production, thus causing considerable economic losses [5]. The disease is caused by the oomycete *Phytophthora infestans* (Mont.) de Bary, one of the most destructive diseases affecting tomato crops [6,7]. The pathogen’s life cycle can be completed in 3–4 days, and rapid inoculum build-up commonly occurs in fields during favorable weather (average temperature of between 10 and 20 °C and high relative humidity of more than 90% or in rainy weather, which leads to a high rate of epidemics. Under such conditions, protective and curative fungicide applications are essential to prevent damage by the pathogen. Late blight tomato fungal disease can be controlled by chemical fungicides, however, there is increasing international concern over the heavy use and application of these products on crops due to their harmful effects on human and environmental health. Additionally, chemical fungicides are responsible for the emergence of pathogen resistance [8,9,10]. Consideration of these risks redirects plant protection strategies towards the use of biological methods in line with sustainable agriculture and the preservation of natural resources [11]. Numerous studies have been conducted on the antagonistic effect of rhizosphere bacteria and their mechanisms of action. Competition for nutrients, space or antibiotic production are the key mechanisms [12,13]. *Bacillus* and *Pseudomonas* species have shown significant promise in inhibiting infections such as *P. infestans* [14,15,16,17,18]. Numerous investigations have shown that bacteria from the genus *Bacillus* are effective against *P. infestans* [16,19]. These bacteria have also been found to promote plant growth [20]. Indeed, the isolation and selection of bacteria with biopesticide potential from the same environment as the cultivated plants can be an alternative that meets all the constraints mentioned above. Additionally, this local selection is better adapted to local conditions and poses a lower ecological or agronomic risk compared to the introduction of foreign strains. The present study aims to investigate the effect of rhizospheric bacteria on late blight tomato fungal disease. This study is part of an approach to sustainable agriculture through the search for rhizospheric bacteria with high pesticide potential in the agroecological management of the main fungi that attack tomato cultivars in the Souss region of Morocco.

## 2. Materials and Methods

### 2.1. Sample Collection

Soil samples were collected from a tomato greenhouse in Douar Ifriane, a rural commune in Inchaden, Chtouka Ait Baha province, located in the Souss-Massa region of Morocco (30°08′53.5″ N, 9°36′42.4″ W). The greenhouse employs integrated crop management practices, including the use of mulch at the base of the plants. A 4.7 cm diameter soil probe was inserted vertically (15–25 cm depth) to collect rhizospheric soil samples. A total of ten samples were collected using the zig-zag sampling method, and approximately 500 g of roots and adhering soil were taken from each plant. The samples were placed in sterile plastic bags, transported to the laboratory, and stored at 4 °C until further analysis.

### 2.2. Isolation and Purification of Bacterial Isolates

Bacterial strains colonizing the rhizosphere (RS), rhizoplane (RP), and endorhizosphere (ES) were isolated under sterile conditions using the suspension dilution method [21]. To isolate bacteria from the rhizosphere (RS), roots were carefully shaken, then 10 g of rhizosphere soil was added to 90 mL of sterile distilled water, and the mixture was vortexed for 2 min to obtain an RS suspension. To isolate bacteria from the rhizoplane (RP), 10 g of previously shaken roots were vortexed for 2 min in 90 mL of sterile distilled water to obtain an RP suspension. To isolate bacteria from the endorhizosphere (ES), the surface of root segments was disinfected with 2.5% sodium hypochlorite solution for 3 min and rinsed three times with sterile distilled water. Ten grams of disinfected roots were ground in a sterile mortar with 90 mL of sterile distilled water to obtain an ES suspension. Suspension samples (RS, RP, and ES) were serially diluted (10^−1^ to 10^−6^) and inoculated separately into a Petri dish containing nutrient agar medium supplemented with cycloheximide (0.1 g/L). After 48 h of incubation at 28 °C, each colony was subcultured and inoculated separately into a Petri dish with fresh nutrient agar and incubated for 48 h at 28 °C. Three replicates were performed per dilution [21,22,23]. Purified isolates were subcultured twice before storage on yeast dextrose carbonate agar (YDC) at 4 °C and −80 °C in 40% glycerol.

### 2.3. Isolation of P. infestans

*P. infestans* was isolated from small shoots exhibiting symptoms of late blight disease. The shoots were disinfected with 75% ethanol and then sliced into 5 mm sections. Subsequently, the sections were placed onto Petri dishes containing previously sterilized potato slices, which were then incubated at 18 °C. Following a five-day incubation period, the mycelium was transferred to V8 medium and incubated in the dark at 20 °C [24,25]. The fungal colony was purified using the hyphal tip method. A 5 mm diameter disc was extracted from the colony’s marginal section, placed in the center of a new Petri dish with V8, and incubated at 20 °C for several days. This process was repeated numerous times until the desired level of purification was achieved. Following purification, the pathogenic fungi were stored at 4 °C. Identification of *P. infestans* was conducted based on morphological and microscopic characteristics. This entailed the determination of the cultural characteristics of the fungal colony, including its appearance, shape, and color. Subsequently, the morphology of the sporangia and sporangiophores was observed using an optical microscope (400×) (BEL Photonics, Model BIO1, Monza, Italy) with Cotton Blue staining.

### 2.4. Pathogenicity Testing

A pathogenicity test was conducted to confirm the virulence of the isolated fungi. Fungal spores were collected by adding 10 mL of sterile water to the culture and filtering the resulting suspension through two layers of cheesecloth to remove the mycelium. The spore concentration was adjusted to 1 × 10^−6^ spores/mL using a hemocytometer. The detached leaves were subjected to a 60-s wash under running tap water, followed by a three-minute surface sterilization with 70% ethanol and a subsequent three-minute treatment with a 1% sodium hypochlorite solution. The samples were then rinsed three times for two minutes each in sterile distilled water, dried with sterile tissue paper, and subsequently air-dried. Inoculation was conducted by placing the surface-sterilized leaves in Petri dishes with tissue paper and spraying them with sterile water to maintain humidity. The samples were inoculated using the wound/drop inoculation method, whereby 5 µL of the conidia suspension is inoculated onto the wound. The control leaves were inoculated with 5 µL of sterile distilled water. The inoculated samples were incubated at 20 °C with a 12-h light/dark cycle [26].

### 2.5. In Vitro Antagonistic Effect of Bacterial Isolates Against P. infestans

A 5 mm-diameter mycelial disc from a 7-day-old culture of the pathogenic fungus was placed in the center of a Petri dish containing PDA medium. A heavy inoculum of the bacterial isolate was inoculated in 1.5 cm strips at the 3 edges of the Petri dish, 2.5 cm from the fungus. The experiment was conducted in triplicate for each isolate [27,28]. The Petri dishes were incubated in the dark for 5 to 7 days at 20 °C [29], after which the mycelial growth of the fungi was measured, and the percentage of inhibition of mycelial growth (PIMG) calculated using the following formula [30]:PIMG = [(r1 − r2)/r1] × 100
where r1 is the radial growth of the fungus in the control, and r2 is the radial growth of the fungus grown in direct confrontation with the bacterial isolate.

### 2.6. Antagonism Mechanism

The isolates that showed inhibitory activity against *P. infestans* were analyzed to assess their capacity to produce hydrogen cyanide and exhibit lipolytic, proteolytic, chitinolytic, cellulase, pectinase, and glucanase activities. All tests were performed in triplicate.


**Production of Hydrogen Cyanide (HCN)**


To reveal the production of hydrogen cyanide (HCN) by bacterial isolates, according Lorck’s method [31], Petri dishes containing LPGA medium (7 g yeast extract, 7 g peptone, 7 g glucose, and 18 g agar per liter), supplemented with 4.4 g/L glycine, were prepared and inoculated with bacterial isolates. A piece of Whatman paper soaked in 5% picric acid and 2% sodium carbonate solution was then placed on the Petri dish lid. The dishes were sealed with parafilm and incubated at 28 °C for 4 days. The change in color of the Whatman paper from yellow to dark orange indicates the production of HCN by the bacteria.


**Lipolytic activity**


Each bacterial isolate was inoculated at three spots on Petri dishes containing peptone agar medium (10 mL Tween80, 10 g peptone, 5 g NaCl, 0.1 g CaCl_2_, and 15 g agar). The dishes were then incubated for 72 h at 28 °C. The appearance of hydrolysis halos around the bacterial inoculum indicates lipolytic activity [32,33].


**Proteolytic activity**


Each bacterial isolate was inoculated at three spots on Petri dishes containing milk agar medium (100 g skimmed milk powder, 5 g peptone, and 15 g agar). The dishes were then incubated for 48 h at 28 °C. The appearance of hydrolysis halos around the bacterial spots indicates proteolytic activity [34].


**Chitinolytic activity**


Bacteria were inoculated on colloidal chitin agar medium (0.07% K_2_HPO_4_, 0.05% MgSO_4_, 0.03% KH_2_PO_4_, 0.001% FeSO_4_, 2% colloidal chitin, and 1.5% agar). After incubation at 30 °C for 5 days, the plates were flooded with Congo red solution (0.03%). Enzymatic activity was manifested by the appearance of clear zones around the colonies [35,36].


**Cellulase production**


Cellulase production was revealed by culturing bacteria on carboxymethyl cellulose agar, which consists of 10 g of carboxymethyl cellulose (sodium salt), 1 g K_2_HPO_4_, 0.2 g MgSO_4_·7H_2_O, 1 g NH_4_NO_3_, 0.05 g FeCl_3_·6H_2_O, 0.02 g CaCl_2_, and 20 g agar [37,38]. The CMC agar plates were inoculated using the spot inoculation method and incubated at 28–30 °C for 4–7 days. Cellulase activity was visualized by staining with 0.1% Congo red solution followed by destaining with 1 M NaCl solution. The formation of hydrolysis halos around the bacterial inoculum indicated cellulase production.


**Pectinase activity**


To reveal the production of pectinase, the bacterial isolate was inoculated in nutrient agar medium supplemented with 5 g/L of apple pectin [38,39]. After incubation for 4–7 days, the Petri dishes were flooded with 1% Lugol’s solution (10 g KI, 5 g iodine, and 100 mL distilled H_2_O). The appearance of a clear zone around the bacterial colonies indicated enzyme activity.


**Glucanase activity**


For glucanase activity, Petri dishes containing nutrient agar medium (1.3 g K_2_HPO_4_, 5 g KH_2_PO_4_, 1 g (NH_4_)_2_SO_4_, 5 g NaCl, 0.24 g/L MgSO_4_·7H_2_O, 3 g yeast extract, and 15 g agar) supplemented with 1% β-glucan [38,40] was inoculated with bacterial isolates. The plates were then incubated at 28–30 °C for 4–7 days to allow for the detection of glucanase activity [41].

### 2.7. In Vivo Study of the Antagonistic Effect of Bacterial Isolates Against P. infestans

To evaluate the effect of the bacterial isolate on *P. infestans* under greenhouse conditions, experiments were conducted using tomato plants (Edmundo cultivar) cultivated in plastic pots containing a 2:1 mixture of peat and perlite in a greenhouse with temperatures between 18 and 24 °C. Once the plants reached the stage of four fully expanded leaves, they were treated with 15 mL of the bacterial suspension with a concentration of 1 × 10⁸ CFU/mL. The suspension was prepared in LB broth medium, incubated overnight at 37 °C, and subsequently applied to the foliage of each plant; the control plants received sterile LB broth. Three days following the administration of the treatment, the plants were inoculated with *P. infestans* (1 × 10⁶ spores/mL) via injection of a 20 μL droplet of the spore suspension at two points on the adaxial surface of each leaf using a sterile 20G syringe. Following inoculation, all plants were incubated for seven days in a growth chamber under controlled conditions (14-h photoperiod, 22 °C, 90% relative humidity) [42].

Disease incidence was determined by counting the number of rotted lesions per plant. Disease severity was evaluated using a standardized severity scale: **0** for no coverage (0%), **1** for 1–25% coverage, **2** for 26–50% coverage, **3** for 51–75% coverage, and **5** for 76–100% coverage. The disease index (**I**) was calculated as a percentage using the formula [43]:I = (Ni/Nt) × 100

Ni is the number of infected leaves by blight, and Nt is the number of plant leaves

### 2.8. Molecular Identification of Strains with High Pesticide Potential

The isolates were identified using conventional PCR (rRNA 16S gene and ITS region) with the Biosystems VERITY™ 2990211635 (Applied Biosystems, Waltham, MA, USA). at the Molecular Analysis Laboratory of the National Center for Scientific and Technical Research CNRST in Rabat.

DNA was extracted using the automated MagPurix Bacterial DNA Extraction Kit (Zinexts Life Science Corp., New Taipei City, Taiwan). The quantity of extracted DNA was determined using a NanoDrop 8000 spectrophotometer (Thermo Fisher Scientific, Waltham, MA, USA). The DNA was then stored at −20 °C until further use.

Polymerase chain reaction (PCR) was conducted on the bacterial DNA samples to obtain a 1500 base pair (bp) amplicon using the universal primers FD1 (5′-AGAGTTTGATCCTGGCTCAG-3′) and RP2 (5′-ACGGCTACCTTGTTACGACTT-3′). DNA polymerase kit (MyTaq) from Bioline (Trento, Italy) was used, along with SeqStudio Flex equipment. Amplification was performed on a thermal Cycler (Thermo Fisher Scientific, Waltham, MA, USA).

PCR products were subjected to electrophoresis on a 1% agarose gel in the presence of a 1 kb molecular weight marker for analysis. Subsequently, PCR products were subjected to sequencing using the SeqStudio FLEX sequencer (Thermo Fisher Scientific, Waltham, MA, USA). Moreover, the obtained sequence was submitted to GenBank and used for BLAST searching against the GenBank database (https://blast.ncbi.nlm.nih.gov/Blast.cgi, accessed on 3 March 2025). Therefore, to characterize each isolate more precisely, the most closely related sequences were used to reconstruct a phylogenetic tree using the UPGMA-analysis approach with Bionumerics software v.7.6 analysis.

### 2.9. Statistical Analysis

Data were analyzed using IBM SPSS Statistics 21 and Microsoft Excel 2016. Results are expressed as mean ± standard deviation (SD) from at least three replicates. A one-way ANOVA was used to determine statistical significance, followed by a Student’s *t*-test (*p* < 0.05) for group comparisons.

## 3. Results

### 3.1. Isolation and Purification of Bacterial Strains from the Rhizosphere of Tomato Crops

A total of 30 bacterial isolates were obtained from the rhizosphere of tomatoes: 15 isolates were obtained from the rhizosphere (RS), four from the endorhizosphere (ES), and 11 from the rhizoplane (RP) (Table 1).

The results of the isolation process demonstrated that the rhizospheric soil of the tomato plant is characterized by high microbial diversity, predominantly bacterial. The data revealed a notable abundance of isolates within the rhizospheric soil (RS) fraction, followed by the rhizoplane (RP) and the endorrhizospheric (ES) compartment. These isolates were subjected to multiple purification cycles to achieve a high level of purity and were then stored at −20 °C in the microbiology laboratory.

### 3.2. Isolation and Pathogenicity of P. infestans

Isolation of *P. infestans*, the causal agent of late blight in tomatoes, is a fundamental step in the study of its pathogenicity and subsequent diagnostic and management practices. Observation of mycelial growth from infected tissues facilitates the isolation process. The second method entailed the direct isolation of the mycelium onto a V8 agar medium (Figure 1), a nutrient-rich medium that is conducive to the growth of *P. infestans*. Following the successful isolation of the colonies, their identification was based on their distinctive morphology as observed on the V8 medium. A microscopic examination served to corroborate the identification of *P. infestans*. Microscopic observation with cotton blue stain (magnification ×1000) revealed the presence of typical structures, including sporangia and sporangiophores. Open sporangia were also observed; this is a crucial aspect of *P. infestans’* life cycle and its capacity to disperse spores, thereby contributing to the dissemination of the disease.

The analysis of *P. infestans* pathogenicity demonstrated that the fungus caused significant lesions on tomato leaves. The initial symptoms manifested as yellow spots that subsequently turned brown, leading to leaf wilting (Figure 2). Additionally, the affected regions exhibited a white, downy layer, which is a distinctive trait of the *P. infestans* mycelium.

### 3.3. Screening of Potential Antagonists Against P. infestans

A total of 30 bacteria, isolated from disparate regions of the rhizospheric soil, were subjected to an assessment of their capacity to inhibit the mycelial growth of *P. infestans*. The findings indicate that the isolates exhibited a degree of inhibition ranging from 11% to 78% for RP3 and RS65, respectively (Figure 3).

Among the 30 isolates, seven (RS65, RP6, RS47, RS46, RP2, RS61, and P30) exhibited significant inhibition (*p* < 0.001), with inhibition rates exceeding 65%. Isolate RS65 showed the highest inhibition rate, at 78.48% (Figure 3). This inhibition was manifested by a reduction of mycelial growth of the pathogen treated with bacterial isolates compared to the control (Figure 4).

The results showed that the seven isolates exhibited significant enzymatic activity (Table 2). Isolate RS61 demonstrated activity for all the targeted enzymes, although it produced a low amount of HCN; in contrast, the other isolates lacked activity for at least one enzyme (Figure 5).

### 3.4. In Vivo Evaluation of the Antifungal Activity of RS65 Against P. infestans

#### Disease Incidence and Severity

Observations were conducted to assess disease incidence across different treatments and then compared to the control. On the 34th day, disease incidence in the control block was recorded at 70.35%, while the treatment exhibited a rate of 54.49%. However, statistical analysis revealed no significant difference in disease incidence between the treatments (Figure 6).

Regarding disease severity, ANOVA results indicated significant differences in late blight severity among the treatments. Statistical analysis revealed a significant difference between the control and the RS65 treatment (Figure 7). The control group showed a severity of 31.26%, while the lowest severity, 16.54%, was observed with the RS65 treatment. This finding demonstrates that the *Bacillus* RS65 treatment provided the significant protection effect against *P. infestans* infection in tomato plants.

### 3.5. Molecular Identification of RS65 Strain

RS65 clusters with several strains of *Bacillus* (Figure 8), showing significant genetic proximity that is supported by high bootstrap values (99% *B. velezensis*, 98% *B. siamensis*, 97% *B. subtilis*, and 91% *B. amyloliquefaciens*), which indicates strong confidence in these phylogenetic relationships. The sequence was submitted to the GenBank adapted reference database under accession numbers PV208381 and was identified as *B. velezensis.* The presence of *Bacillus amyloliquefaciens* and *Bacillus velezensis* in close proximity to RS65 suggests potential phenotypic similarities, particularly in the production of enzymes and secondary metabolites with biopesticide activity.

## 4. Discussion

Plant fungal diseases pose a major threat to the yields and productivity of economically important crops worldwide. Common management strategies include the use of resistant cultivars, cultural practices, and chemical applications. However, biopesticides currently account for only 2% of plant protectants globally, although their use is increasing by approximately 10% annually. Global biopesticide production is estimated to be over 3000 tons per year, with nearly 90% derived from the microbial agent *Bacillus thuringiensis* [44,45]. However, this dominance also highlights the need to diversify microbial agents in the market, particularly by seeking more bacteria that play a crucial role in combating fungal diseases. Diversifying biopesticide strategies could thus enhance the management of phytopathogenic threats across various contexts and address more specific needs in crop protection. The results of this study demonstrated that bacterial cultures can control the effect of *P. infestans* through their antagonistic properties. A substantial body of evidence indicates that the rhizosphere soil of a wide range of plants is predominantly populated by bacteria, particularly members of the *Bacillus* genus [46,47]. Bacterial genera such as *Bacillus*, *Pseudomonas*, and *Enterobacter*, fungi belonging to *Pythium* and *Trichoderma* genus, and *Actinomycetes* have been identified as having notable biocontrol potential [3,48,49,50,51]. Our results demonstrated that this bacterial genus exhibits considerable potential for the protection of plants against pathogenic diseases. The findings of this study corroborate those of previous research, indicating the efficacy of *Bacillus* bacteria as a suitable biocontrol agent against late blight; the results of this study, based on the analysis of 30 strains isolated from the rhizosphere of the tomato plant, suggest that applying the biocontrol agent *Bacillus* RS65 could effectively mitigate the impact of late blight in tomato plants, offering a promising alternative in sustainable disease management.

Phylogenetic analysis identified RS65 as a member of the *Bacillus* genus, closely related to *B. amyloliquefaciens* and *B. velezensis*. These species are recognized for their biopesticide properties [52,53], suggesting that RS65 may exhibit similar traits. These findings support the hypothesis that RS65 could be used in an integrated disease control program for sustainable management of tomato late blight as an effective biocontrol agent against fungal phytopathogens in tomatoes, specifically targeting the pathogens responsible for late blight disease. Additionally, these species are known to produce a diverse range of bioactive compounds [54] and may also exhibit promising antifungal properties. The direct mechanisms of bacterial biocontrol have been reported to include the following: antagonistic action, the production of antibiotic compounds, competition for nutrients, and siderophore-mediated competition for iron and/or the production of extracellular enzymes [51,55,56,57]. The current study’s results demonstrated that all strains tested produced the HCN and lytic enzymes, including proteases, cellulases, and chitinases, with variations in enzymatic activity. This variability underscores the potential of these isolates to act synergistically; for instance, combining ES1 (high HCN production), RP2 (notable production of chitinase, pectinase, cellulase, and glucanase), and RS46 (high production of lipase and protease) could offer significant advantages (Figure 5). Similarly, the synergy between RS65, a rhizospheric strain, and RP6, a rhizoplane-associated strain, could further enhance biocontrol efficacy, leveraging their complementary traits to target fungal phytopathogens effectively.

The high enzymatic activity of *Bacillus* species may play a role in the induction of the natural defenses of the infected plant through the phenomenon of elicitation, which results in the expression of defense genes. Ramette et al. [58] reported that the microbial production of HCN is an important antifungal trait in the control of root-infecting fungi. In a similar context, Kumari and Khanna [59] reported that the plant growth-promoting rhizobacteria isolate (15B) significantly inhibited the growth of *F. oxysporum f.sp. ciceri* by producing volatile organic compounds (VOCs), resulting in 64.2% inhibition compared to controls. Certain enzyme-producing bacteria are capable of destroying oospores of phytopathogenic fungi [60] and influencing the spore germination and germ-tube elongation of phytopathogenic fungi [61,62]. The success of *Bacillus* species as biocontrol agents can be attributed to the production of a diverse range of peptide antibiotics, including iturin A, mycobacillin, subtilin, and bacilysin, as well as 25 different basic chemical structures with proven antifungal secondary metabolites [63,64]. A significant degree of variation is observed in the quantity and diversity of antifungal compounds produced by *B. subtilis* [65]. A combination of different mechanisms plays an important role in the inhibition of *P. infestans in vitro* and under greenhouse conditions, affecting fruits and leaves. The potential production of volatile and diffusible antagonistic metabolites suggests that selected bacteria may act as antagonists against various phytopathogenic fungi that infect tomatoes and other crops. The findings of this study indicate that RS65 has the potential to serve as a valuable bio-inoculant for the advancement of sustainable agricultural practices in agrosystems, particularly in the context of managing *P. infestans*; furthermore, it has the potential to serve as a protective biopesticide, offering a viable alternative to chemical fungicides in the management of fungal diseases. RS65 can be used in an integrated control program for tomato late blight.

### Perspectives

Future research should focus on the biochemical and molecular characterization of the bioactive compounds produced by RS65 to better understand its mechanisms of action. Quantitative and qualitative assessments of its antifungal metabolites are crucial for optimizing its application. Additionally, field trials should be conducted to evaluate the efficacy of RS65 under different environmental conditions and cropping systems. Exploring its use in biofertilizer formulations or as part of integrated control programs would further enhance its role in sustainable agriculture.

## Figures and Tables

**Figure 1 microorganisms-13-00656-f001:**
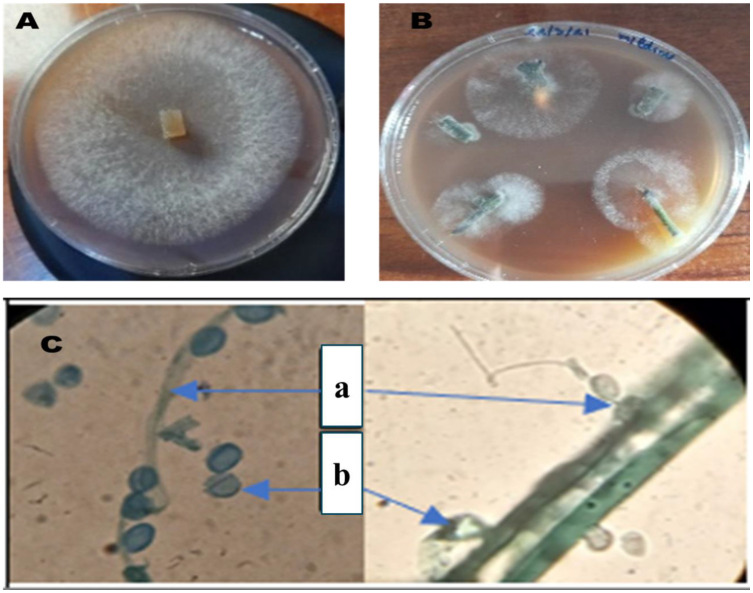
*P. infestans* colony on V8 medium (**A**); isolation of mycelium directly on V8 medium (**B**); microscopic observation of *P. infestans* (×1000) using cotton blue (**a**: sporangia and sporangiophore, **b**: open sporangia) (**C**).

**Figure 2 microorganisms-13-00656-f002:**
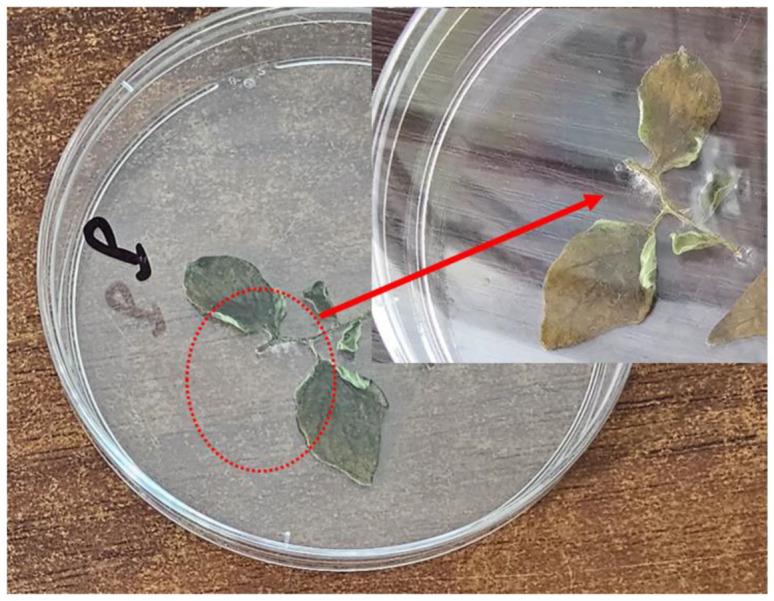
Pathogenicity of *P. infestans*, causing significant lesions on tomato leaves characterized by oily spots and leaf browning.

**Figure 3 microorganisms-13-00656-f003:**
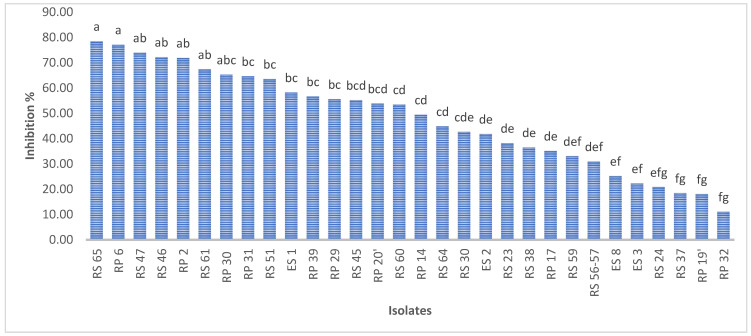
Percentage inhibition of *P. infestans* mycelial growth by bacterial isolates. Bars with the same letters are not significantly different at *p* < 0.001 using student’s *t*-test.

**Figure 4 microorganisms-13-00656-f004:**
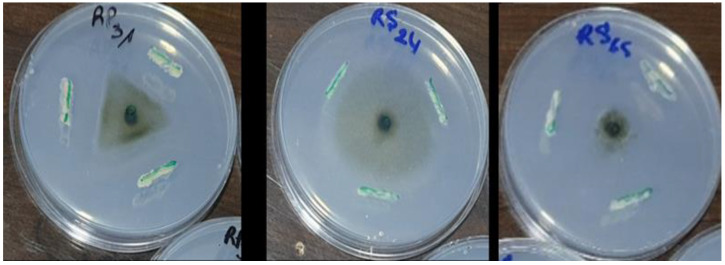
Effect of bacterial isolates on mycelial growth of *P. infestans*.

**Figure 5 microorganisms-13-00656-f005:**
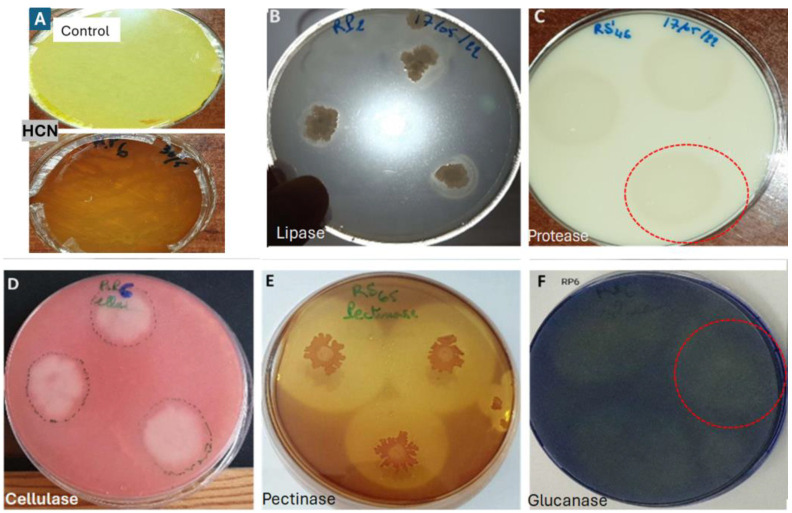
Manifestation of enzymatic activity of bacterial isolates, (**A**) HCN production; (**B**) lipolytic activity; (**C**) proteolytic activity; (**D**) cellulase production; (**E**) pectinase production; (**F**) Glucanase production.

**Figure 6 microorganisms-13-00656-f006:**
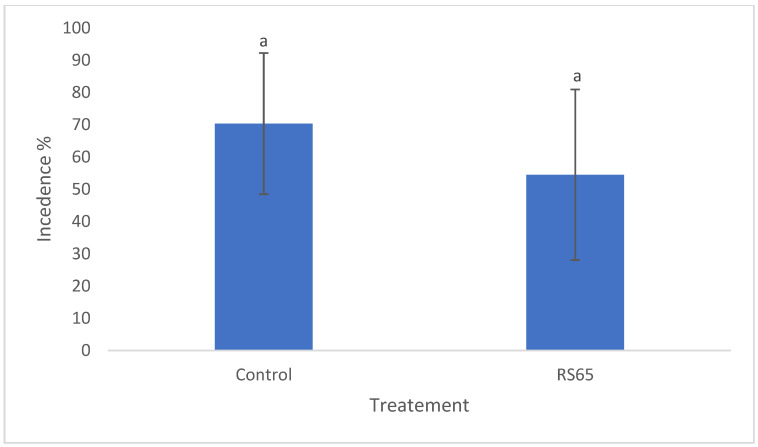
Comparison of disease incidence in tomato plants treated with RS65 versus untreated controls under greenhouse conditions. Bars with the same letters are not significantly different at *p* < 0.05 using student’s *t*-test.

**Figure 7 microorganisms-13-00656-f007:**
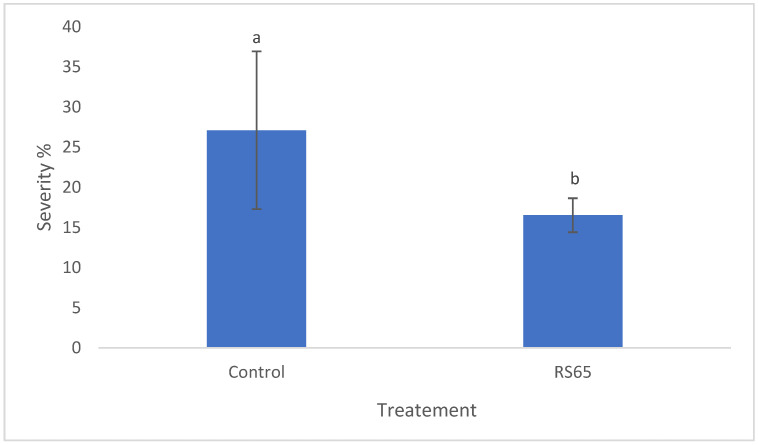
Effect of RS65 isolate on the severity of *P. infestans* infection in tomato plants under greenhouse conditions. Bars with the same letters are not significantly different at *p* < 0.05 using student’s *t*-test.

**Figure 8 microorganisms-13-00656-f008:**
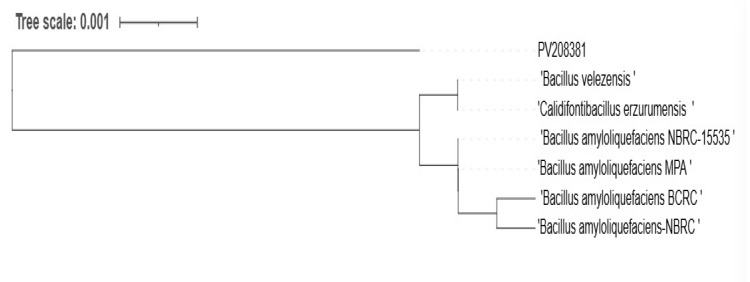
Phylogenetic tree of the RS65 strain (PV208381) constructed based on an alignment of partial sequences of the 16S ribosomal RNA gene (PCF, PDR, PAF, and PEF) using the NCBI platform, applying the neighbor-joining method with bootstrap support.

**Table 1 microorganisms-13-00656-t001:** Isolation of bacterial isolates from the rhizospheric soil of tomato plants.

	Rhizospheric Soil (RS)	Endorrhizospheric Soil (ES)	Rhizoplan (RP)
Isolates	15 isolates: RS65; RS47; RS46; RS61; RS51; RS45; RS60; RS64; RS30; RS23; RS38; RS59; RS56-57; RS24; RS37;	4 isolates: ES1; ES2; ES8; ES3	11 isolates: RP6; RP2; RP30; RP31; RP39; RP29; RP20′; RP14; RP17; RP19′; RP32

**Table 2 microorganisms-13-00656-t002:** Enzymatic activity of the seven selected isolates.

	Bacterial Isolate						
	RS61	RP6	RS65	RP2	RS46	RS47	RP30
Production of cyanide HCN	+	++	++	-	-	-	++
lipolytic activity	++	-	-	+	++	-	-
Proteolytic activity	++	++	++	+	++	-	-
Chitinolytic activity	+++	+	+	+++	+	+	+++
Cellulase production	+++	++	++	+++	++	+	+++
Pectinase production	+++	+++	+++	+++	-	+	-
Glucanase production	++	+++	++	+++	+	-	-

+ low production, ++ medium production, +++ high production, - no production: The level of HCN production is indicated by the color intensity of the filter paper, which changes from yellow to brown. No production is indicated by the yellow color (denoted by (-)), low production by the orange color (denoted by (+)), high production by the brown color (denoted by (+++)), and intermediate production by the intermediate color (denoted by (++)). For enzymatic activity, the level of production is indicated by the diameter halo around the colony.

## Data Availability

The original contributions presented in this study are included in the article. Further inquiries can be directed to the corresponding authors.

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
