# Peer review of "Biocontrol Potential of Bacillus velezensis RS65 Against Phytophthora infestans: A Sustainable Strategy for Managing Tomato Late Blight"

_microorganisms, 2025, doi:10.3390/microorganisms13030656_

Round 1

Reviewer 1 Report

Comments and Suggestions for Authors

Review on “The effect of rhizospheric Bacillus on the tomato late blight pathogen (Phytophthora infestans)” for manuscript ID microorganisms-3488376

In this manuscript the authors describe the effect of rhizospheric bacteria on late blight tomato fungal disease. The introduction section lacks the current state of knowledge about Bacillus bacteria application against the Phytophthora infestans. Please extend the Introduction with the review of the recent studies on the topic.

My questions and comments:

L34: ref. [1] is invalid

L58: “The present study aims to … to elucidate the mechanism of their antagonistic activity.” – no mechanism is suggested in Results/Discussion.

L69-70: It is not clear how this information can be useful for the reader to understand the topic of the study.

L71-76 Please explain how you collected samples, as it is not entirely clear from the text. So, first you selected random plants in order to take samples of the rhizosphere with a probe. Explain: in order to extract the root system, were the same or different tomato plants selected? What is the diameter of the probe?

L110-111 Is it appropriate to use the term "conidia" or "conidiophores" in relation to oomycetes, which I usually use for asomycetes?

L112 Please explain what SDV means.

L169 0.03% KHâ‚‚POâ‚„ is repeated twice

L156, 161, 166: References are usually added when describing a particular method.

L229: It is a good practice to submit the sequences to NCBI GenBank for further use and citation

L239: Table 1: “isolation of isolates from rhizospheric parts of tomato crops” – please rephrase

L240-242: It is not clear which isolates the authors have in mind. If the authors list only those isolates that showed activity against late blight, this should be stated more clearly.

Ref. [6, 7] are non-actual, it’s better to refer to more recent data.

Figure 2 is missing.

Figure 4: the image scale does not allow the antagonism effect to be assessed.

Figure 8 is unreadable. The close species are poorly separated using 16S phylogeny only, the multi locus or whole genome phylogeny could be more consistent.

Table 2: what are criteria for low and high production?

Author Response

comments

Answer

Rev 1

The introduction section lacks the current state of knowledge about Bacillus bacteria application against the Phytophthora infestans. Please extend the Introduction with the review of the recent studies on the topic.

The text “Numerous studies have been conducted on the antagonistic effect of rhizosphere bacteria and their mechanisms of action. Competition for nutrients, space or antibiotic production are the key mechanisms[12,13]. Bacillus and Pseudomonas species have shown significant promise in inhibiting infections such as P. infestans[14–18].Numerous investigations have shown that bacteria from the genus Bacillus are effective against P. infestans [16,19,20]. These bacteria have also been found to promote plant growth[21]”  was added into the introduction section .

L34: ref. [1] is invalid

The ref ref. [1] was replaced by Mariz-Ponte et al.,2019

L58: “The present study aims to … to elucidate the mechanism of their antagonistic activity.” – no mechanism is suggested in Results/Discussion.

The paragraph “The present study aims to investigate the effect of rhizospheric bacteria on late blight tomato fungal disease and to elucidate the mechanism of their antagonistic activity” was replaced by “The present study aims to investigate the effect of rhizospheric bacteria on late blight tomato fungal disease”.

L69-70: It is not clear how this information can be useful for the reader to understand the topic of the study.

The text “This was the first crop on a 1.5-hectare wild plot of sandy soil dedicated to the production of organic tomatoes for export” was removed

L71-76 Please explain how you collected samples, as it is not entirely clear from the text. So, first you selected random plants in order to take samples of the rhizosphere with a probe. Explain: in order to extract the root system, were the same or different tomato plants selected? What is the diameter of the probe?

The paragraph “The greenhouse employs integrated crop management practices, including the use of mulch at the base of the plants. A soil probe was carefully inserted vertically into the root zone of healthy tomato plants to a depth of 15–25 cm to minimize root disturbance. The probe was gently rotated to extract rhizospheric soil samples, specifically the soil most strongly adhering to the roots. Simultaneously, root samples were collected to isolate microbial communities from the rhizoplane and endosphere. Approximately 500 g of roots and adhering soil were collected using the zigzag sampling method. In total, ten samples were collected and placed in sterile plastic bags, transported to the laboratory, and stored at 4°C until further analysis.”  Was replaced by  “The greenhouse employs integrated crop management practices, including the use of mulch at the base of the plants. A 4.7 cm diameter soil probe was carefully inserted vertically into the rhizospheric soil zone of the plants (depth 15–25 cm). The probe was gently rotated to extract rhizospheric soil samples. A total of ten samples were collected using the zigzag sampling method, and approximately 500 g of roots and adhering soil were taken from each plant. The samples were placed in sterile plastic bags, transported to the laboratory, and stored at 4°C until further analysis.”

L110-111 Is it appropriate to use the term "conidia" or "conidiophores" in relation to oomycetes, which I usually use for asomycetes?

The paragraph “Subsequently, the morphology of the conidia and conidiophores was observed using an optical microscope (400x) with Cotton Blue staining.” was replaced by  “Subsequently, the morphology of the sporangia and sporangiophores was observed using an optical microscope (400x) with Cotton Blue staining.”

L112 Please explain what SDV means.

The abbreviation “SDW” was replaced by sterile distilled water (L 125)

L169 0.03% KHâ‚‚POâ‚„ is repeated twice

the repetition was deleted

L156, 161, 166: References are usually added when describing a particular method.

the references were added to the text

·       Lorck, H. (1948). Production of hydrocyanic acid by bacteria. Physiologia Plantarum1(2).

·       Veerapagu, M., Narayanan, A. S., Ponmurugan, K., & Jeya, K. R. (2013). Screening selection identification production and optimization of bacterial lipase from oil spilled soil. Asian J Pharm Clin Res6(3), 62-67.

·       Yahia, M., Mohamed, M., Othman, M., Mostafa, D., Gomaa, M., Fahmy, M., ... & Abu-Hussien, S. (2020). Isolation and identification of antibiotic producing Pseudomonas fluorescens NBRC-14160 from Delta Soil in Egypt. Arab Universities Journal of Agricultural Sciences28(3), 797-808.

·       Admassie, M., Woldehawariat, Y., & Alemu, T. (2022). In vitro evaluation of extracellular enzyme activity and its biocontrol efficacy of bacterial isolates from pepper plants for the management of Phytophthora capsici. BioMed research international2022(1), 6778352.

L229: It is a good practice to submit the sequences to NCBI GenBank for further use and citation

 Yes, we will submit the sequences to NCBI GenBank for further use and citation

L239: Table 1: “isolation of isolates from rhizospheric parts of tomato crops” – please rephrase

the table title “Isolation of isolates from rhizosphere parts of tomato crops” was replaced by “Isolation of bacterial isolates from the rhizospheric soil of tomato plants.”

L240-242: It is not clear which isolates the authors have in mind. If the authors list only those isolates that showed activity against late blight, this should be stated more clearly.

The bacterial strains listed in the table are those isolated from the rhizosphere of tomato plants. In this manuscript we have listed only 30 isolates. Their activity against late blight was tested in the Petri plate test. The bacterium RS65, which showed a high inhibition of P. infestans, was tested in vivo.

Ref. [6, 7] are non-actual, it’s better to refer to more recent data.

Changed by :

-        Mazumdar, P.; Singh, P.; Kethiravan, D.; Ramathani, I.; Ramakrishnan, N. Late Blight in Tomato: Insights into the Pathogenesis of the Aggressive Pathogen Phytophthora Infestans and Future Research Priorities. Planta 2021 253:6 2021, 253, 1–24, doi:10.1007/S00425-021-03636-X.

-        Ben Naim, Y.; Cohen, Y. Replacing Mancozeb with Alternative Fungicides for the Control of Late Blight in Potato. Journal of Fungi 2023, Vol. 9, Page 1046 2023, 9, 1046, doi:10.3390/JOF9111046.

Figure 2 is missing.

Added

Figure 4: the image scale does not allow the antagonism effect to be assessed.

The title and image were changed

Figure 8 is unreadable. The close species are poorly separated using 16S phylogeny only, the multi locus or whole genome phylogeny could be more consistent.

We agree that a multi-locus sequence typing (MLST) or whole-genome phylogeny approach would provide a more consistent and accurate separation of these species.

The whole genome sequencing strain RS65 is planned for future studies. This will make it possible to define the genes responsible for the enzyme production and antifungal activity of this strain.

Table 2: what are criteria for low and high production?

We based our assessment on the diameter of the lysis halo in the Petri plate

-          Absence of production

+    Low production,

++  Medium production

+++  High production

The production of HCN was determined by monitoring the change in color of the Whatman paper (the intensity of color) from yellow to dark orange. This change was indicative of HCN production by the bacteria in comparison to the control (yellow color).

Reviewer 2 Report

Comments and Suggestions for Authors

Dear Authors,

I have carefully reviewed the manuscript titled The Effect of Rhizospheric Bacillus on the Tomato Late Blight Pathogen (Phytophthora infestans). This study provides valuable insights into the biocontrol potential of rhizosphere-derived Bacillus isolates against Phytophthora infestans, a major pathogen responsible for late blight in tomatoes. By evaluating 30 bacterial isolates for their antagonistic activity, the research identifies six strains with significant inhibitory effects, particularly isolate RS65, which exhibited the highest inhibition rate of 78.48% in vitro. Furthermore, the study highlights the enzymatic activity and secondary metabolite production of these isolates, suggesting multiple biocontrol mechanisms. Under greenhouse conditions, RS65 treatment significantly reduced disease severity compared to the control, reinforcing its potential as a biological control agent. Molecular identification confirmed RS65 as Bacillus velezensis, further supporting its role in integrated disease management strategies for tomato late blight. This research makes a valuable contribution to sustainable plant protection methods.

To further strengthen its scientific depth and practical applicability, additional refinements are recommended. A more detailed characterization of Bacillus RS65's mode of action through molecular, transcriptomic, or metabolomic analyses would provide deeper insights into the specific pathways involved in its biocontrol activity, such as antifungal metabolite production, enzymatic action, and volatile organic compound release. Additionally, a direct comparison with chemical fungicides and other biocontrol agents, such as Trichoderma spp., would clarify its relative efficacy, potential synergistic effects, and suitability as an alternative to conventional treatments. Investigating the impact of Bacillus RS65 on soil and plant microbiomes is also crucial, as it would reveal its influence on microbial diversity, beneficial interactions, and possible unintended consequences, ensuring its safe and effective application in sustainable agriculture.

Based on the strengths of the study and the suggested refinements, I recommend accepting the manuscript with minor revisions.

Best regards.

Author Response

comments

Answer

Rev 2

I have carefully reviewed the manuscript titled The Effect of Rhizospheric Bacillus on the Tomato Late Blight Pathogen (Phytophthora infestans). This study provides valuable insights into the biocontrol potential of rhizosphere-derived Bacillus isolates against Phytophthora infestans, a major pathogen responsible for late blight in tomatoes. By evaluating 30 bacterial isolates for their antagonistic activity, the research identifies six strains with significant inhibitory effects, particularly isolate RS65, which exhibited the highest inhibition rate of 78.48% in vitro. Furthermore, the study highlights the enzymatic activity and secondary metabolite production of these isolates, suggesting multiple biocontrol mechanisms. Under greenhouse conditions, RS65 treatment significantly reduced disease severity compared to the control, reinforcing its potential as a biological control agent. Molecular identification confirmed RS65 as Bacillus velezensis, further supporting its role in integrated disease management strategies for tomato late blight. This research makes a valuable contribution to sustainable plant protection methods.

To further strengthen its scientific depth and practical applicability, additional refinements are recommended. A more detailed characterization of Bacillus RS65's mode of action through molecular, transcriptomic, or metabolomic analyses would provide deeper insights into the specific pathways involved in its biocontrol activity, such as antifungal metabolite production, enzymatic action, and volatile organic compound release. Additionally, a direct comparison with chemical fungicides and other biocontrol agents, such as Trichoderma spp., would clarify its relative efficacy, potential synergistic effects, and suitability as an alternative to conventional treatments. Investigating the impact of Bacillus RS65 on soil and plant microbiomes is also crucial, as it would reveal its influence on microbial diversity, beneficial interactions, and possible unintended consequences, ensuring its safe and effective application in sustainable agriculture.

Based on the strengths of the study and the suggested refinements, I recommend accepting the manuscript with minor revisions.

-Thank you very much for your valuable scientific critique, your encouragement, and your guidance regarding the second stage of this research, which will focus on three main axes:

·       Characterization of Bacillus RS65's mode of action through molecular, transcriptomic, or metabolomic analyses.

·       A direct comparison with chemical fungicides and other biocontrol agents.

Investigating the impact of Bacillus RS65 on soil and plant microbiomes.

-Many improvement and modifications were made in the text  ( highlighted ), references , and  figures

Reviewer 3 Report

Comments and Suggestions for Authors

The manuscript, titled "The effect of rhizospheric Bacillus on the tomato late blight pathogen (Phytophthora infestans)", presents the initial steps in the development of a bio-pesticide. One of the most promising areas of bio-fungicide developments is symbiotic bacteria of the genus Bacillus isolated from the rhizosphere or from asymptomatic (healthy) plants. In the present study, such isolations were made from the rhizosphere (RS), rhizoplane (RP), and endorhizosphere (ES) of greenhouse tomato plants. Following successful isolations, the in vitro efficacy of each bacterial strain against the target pathogenic fungus, as well as its ability to produce various enzymes, were determined. The most effective strain in the laboratory (RS65) was also tested for in vivo antagonism. Molecular biology studies were also used to determine the genetic similarity of each bacterial strain to identified Bacillus species.

The introduction is sufficient, but I recommend a more detailed overview of bacterial-based biopesticides effective against Phytophthora fungal species.

The materials and methods section is elaborated in sufficient detail and the descriptions allow the reproduction of the performed studies.

The presentation of the results is basically adequate, but suffers from several shortcomings, which I suggest to improve:
- in line 212 of the manuscript, I suggest to replace the word "mildew" with the term "blight".
- in the presentation of Results section, the title of Figure 2 is included, but the figure itself is missing. I suggest to add it!
- in the case of Figure 4, the image is too small. In order to make it clearer, I suggest to enlarge it.
- in the legend of Table 2, the term "++-medium activity" is missing
- in the case of Figure 5, the explanation of the subfigure F is missing. In the same figure, the subfigure A is too small, not suitable for presentation. I suggest to present a photo of a single Petri dish here as well, similarly to the other subfigures!

  • in line 319, I suggest using the term "significant protection effect" instead of the term "best protection", since in this study only one bacterial strain was tested in addition to the untreated control.
    - in line 328, the listing is not uniform. I suggest using B. subtilis in the listing instead of the incorrect name B. Bacillus subtilis.

The conclusion chapter partly analyzes the achieved results well. At the same time, I recommend reviewing that the most effective antagonist bacterial strain in in vitro efficacy studies may not be the most effective in in vivo studies, especially in inducing the elicitor effect in cultivated plants. Antagonists can achieve the desired protective effect against the targeted pathogenic fungus by several (often synergistic) bio-mechanisms of action. Based on these, it may be recommended to associate different bacterial strains to achieve the required level of efficacy.

After implementing the corrections suggested above, I recommend resubmitting the manuscript.

Author Response

comments

Answer

Rev 3

The introduction is sufficient, but I recommend a more detailed overview of bacterial-based biopesticides effective against Phytophthora fungal species.

 The paragraph “Numerous studies have been conducted on the antagonistic effect of rhizosphere bacteria and their mechanisms of action. Competition for nutrients, space or antibiotic production are the key mechanisms[12,13]. Bacillus and Pseudomonas species have shown significant promise in inhibiting infections such as P. infestans[14–18].Numerous investigations have shown that bacteria from the genus Bacillus are effective against P. infestans [16,19,20]. These bacteria have also been found to promote plant growth[21]” was added  to the maniscript 

in line 212 of the manuscript, I suggest to replace the word "mildew" with the term "blight".

Made in the text

- in the presentation of Results section, the title of Figure 2 is included, but the figure itself is missing. I suggest to add it!

 The Figure 2 was added in the maniscript

- in the case of Figure 4, the image is too small. In order to make it clearer, I suggest to enlarge it.

The title and the image were changed in the maniscript

- in the legend of Table 2, the term "++-medium activity" is missing

Made in the text

- in the case of Figure 5, the explanation of the subfigure F is missing. In the same figure, the subfigure A is too small, not suitable for presentation. I suggest to present a photo of a single Petri dish here as well, similarly to the other subfigures!

The image was adjusted

in line 319, I suggest using the term "significant protection effect" instead of the term "best protection", since in this study only one bacterial strain was tested in addition to the untreated control.

 The term was changed

- in line 328, the listing is not uniform. I suggest using B. subtilis in the listing instead of the incorrect name B. Bacillus subtilis.

Made in the text

The conclusion chapter partly analyzes the achieved results well. At the same time, I recommend reviewing that the most effective antagonist bacterial strain in in vitro efficacy studies may not be the most effective in in vivo studies, especially in inducing the elicitor effect in cultivated plants. Antagonists can achieve the desired protective effect against the targeted pathogenic fungus by several (often synergistic) bio-mechanisms of action. Based on these, it may be recommended to associate different bacterial strains to achieve the required level of efficacy.

Thank you very much for your valuable feedback and suggestions.

Yes, it's possible that the most effective antagonist bacterial strain in in vitro tests is not always the most effective in in vivo test. In this paper, we only tested the RS65 strain in vivo, which showed a high antagonist effect in Petri plates.

 For the next work, we planned to test other isolates and their consortium to achieve the desired protective effect against the target pathogen through synergistic bio-mechanisms of action.

Round 2

Reviewer 1 Report

Comments and Suggestions for Authors

Review on “The effect of rhizospheric Bacillus on the tomato late blight pathogen (Phytophthora infestans)” for manuscript ID microorganisms-3488376

At first, I would like to thank the authors for the improving the manuscript, but some concerns remain to be addressed.

Major points:

The confidence interval is required at Figures 6 and 7.

The phylogenetic tree has to be revised: the parent Phylum of genus Bacillus is renamed to Bacillota (former Firmicutes, see https://lpsn.dsmz.de/phylum/bacillota). The species/strains names are unclear, the choice of particular species to construct the phylogenetic tree isn’t explained. Inclusion of “uncultured” strains into the phylogenetic tree is questionable.

L229: I strongly recommend to wait until release of NCBI GenBank submission and include the accession numbers into the manuscript.

Minor points:

L93: use superscript to power of 10th.

L66: while only the Bacillus bacteria are considered in the study, the aim could be narrowed to particular genus.

Table 2: Please include the criteria for low and high production into the legend.

Author Response

comments

Answer

Rev 1

The confidence interval is required at Figures 6 and 7.

The confidence interval  was added

The phylogenetic tree has to be revised: the parent Phylum of genus Bacillus is renamed to Bacillota (former Firmicutes, see https://lpsn.dsmz.de/phylum/bacillota). The species/strains names are unclear, the choice of particular species to construct the phylogenetic tree isn’t explained. Inclusion of “uncultured” strains into the phylogenetic tree is questionable.

L229: I strongly recommend to wait until release of NCBI GenBank submission and include the accession numbers into the manuscript.

sequence was submitted to GenBank and used for BLAST searching against the GenBank database (http://www.blast.ncbi.nlm.nih.gov). Therefore, to characterize each isolate more precisely, the most closely related sequences were used to reconstruct a phylogenetic tree using the UPGMA-analysis approach with Bionumerics software analysis.

Phylogenetic tree Was changed in the manuscript

The sequence was submitted to the GenBank adapted reference database under accession numbers OR167382 and was identified as

L93: use superscript to power of 10th.

Made in the manuscript  : (10-1 to 10-6)

L66: while only the Bacillus bacteria are considered in the study, the aim could be narrowed to particular genus.

When we started this study, we didn't focus on the genus Bacillus. Our goal was to find rhizospheric bacteria effective against P infestation. But after we found one type of bacteria, we realized it was Bacillus.

Table 2: Please include the criteria for low and high production into the legend.

Added to the text

Reviewer 3 Report

Comments and Suggestions for Authors

After implementing the suggested corrections, I recommend publishing the manuscript as a scientific article.

Author Response

Comment: After implementing the suggested corrections, I recommend publishing the manuscript as a scientific article.

Answer : Thank you